# Study on the Preparation of Ionic Liquid Doped Chitosan/Cellulose-Based Electroactive Composites

**DOI:** 10.3390/ijms20246198

**Published:** 2019-12-09

**Authors:** Fang Wang, Chong Xie, Liying Qian, Beihai He, Junrong Li

**Affiliations:** State Key Laboratory of Pulp and Paper Engineering, South China University of Technology, Guangzhou 510641, China; fangwangjuly@163.com (F.W.); xiechonggz@outlook.com (C.X.); ppebhhe@scut.edu.cn (B.H.)

**Keywords:** electroactive materials, electroactive properties, chitosan/cellulose composite film, [EMIM] Ac, mechanical properties

## Abstract

Electro-actuated polymer (EAP) can change its shape or volume under the action of an external electric field and shows similar behavioral characteristics with those of biological muscles, and so it has good application prospects in aerospace, bionic robots, and other fields. The properties of cellulose-based electroactive materials are similar to ionic EAP materials, although they have higher Young’s modulus and lower energy consumption. However, cellulose-based electroactive materials have a more obvious deficiency—their actuation performance is often more significantly affected by ambient humidity due to the hygroscopicity caused by the strong hydrophilic structure of cellulose itself. Compared with cellulose, chitosan has good film-forming and water retention properties, and its compatibility with cellulose is very excellent. In this study, a chitosan/cellulose composite film doped with ionic liquid, 1-ethyl-3-methylimidazolium acetate ([EMIM]Ac), was prepared by co-dissolution and regeneration process using [EMIM]Ac as the solvent. After that, a conductive polymer, poly(3,4-ethylenedioxythiophene)/poly (styrene sulfonate) (PEDOT: PSS), was deposited on the surface of the resulted composite, and then a kind of cellulose-based electroactive composites were obtained. The results showed that the end bending deformation amplitude of the resulted material was increased by 2.3 times higher than that of the pure cellulose film under the same conditions, and the maximum deformation amplitude reached 7.3 mm. The tensile strength of the chitosan/cellulose composite film was 53.68% higher than that of the cellulose film, and the Young’s modulus was increased by 72.52%. Furthermore, in comparison with the pure cellulose film, the water retention of the composite film increased and the water absorption rate decreased obviously, which meant that the resistance of the material to changes in environmental humidity was greatly improved.

## 1. Introduction

Smart materials play important roles in the fields of military, construction, and daily life because of their multi-functional and adaptable designs. As one type of smart materials, the electroactive polymer (EAP) can generate deformation and deflection displacement under an electric field, or generate a response charge under stress [1]. These properties widen the potential application of EAP materials in actuators [2,3,4], Micro Electro Mechanical Systems devices [5,6], and biomimetic robots [7,8,9]. According to its actuation mechanism, electroactive polymer can be divided into two categories, i.e., ionic EAP materials related to ion migration and diffusion mechanism, and electronic EAP materials related to electric field driven and Coulomb force, respectively [10,11,12]. Generally, electronic EAP requires a higher actuation voltage, while ionic EAP materials require only a small actuation voltage [13]. However, the electroactive properties of ionic EAP materials are strongly affected by the changes of environmental humidity, so it is difficult to maintain their electroactive behavior enduringly under the condition of DC electric field [14]. 

Cellulose has a physically and chemically highly specific composite structure, and there are oriented uniaxial crystallization regions and amorphous regions in this structure during the biosynthesis processes [15]. In the last century, Eiichi Fukada conducted in-depth research on natural polymer materials with such interesting structures, and found that this structure of the wood can produce piezoelectric effects [10]. The natural self-orientation of cellulose exhibits shear piezoelectricity due to the internal rotation of a polar atomic group attached to a asymmetric carbon atom [10,16]. The piezoelectric effect of cellulose provides guidance and the basis for the research of cellulose materials in the field of electrical actuation. Jaehwan Kim and his collaborators prepared a kind of cellulose-based electroactive material, named EAPap, which can be deflected under the effect of an electric field [17]. In addition, they found that in the crystalline and amorphous regions of cellulose, ions migration with free water under the action of external electric field led to an uneven expansion of the cellulose-based material [18]. Many researchers have used ionic liquids to regenerate and prepare EAPap, and doped with polyvinyl alcohol, polyethylene glycol, and ionic liquids in order to increase their electroactive performance [4,11,19]. Ionic liquid is not volatile and has good affinity with cellulose matrix and residual water, therefore, it helps to reduce the disappearance of a material’s electroactive behavior caused by the loss of water in the cellulose. A cellulose-based material has many advantages, such as being light weight and biodegradable, and having high strength, a low actuation voltage, and large actuation deformation [20,21,22,23]. However, due to the hydrophilic structure of cellulose, the ions in the crystalline and amorphous regions of the cellulose migrated with free water under an applied electrical field, resulting in a non-uniform expansion of the cellulose-based material [18], and so, cellulose-based EAP was sensitive to humidity and temperature, and its driving behavior decayed over time [24].

Chitosan is not only abundant in resources, but also has excellent properties, such as biodegradability, good biocompatibility, antibacterial function, and water retention. Chitosan-based composites can be obtained by utilizing the good film-forming properties of chitosan. Altınkaya prepared a chitosan-based material with good mechanical properties [5]. Chitosan and cellulose have good affinity, and they can be combined in order to obtain good composite materials. By doping chitosan, cellulose-based materials can have stronger mechanical properties [25]. The EAP materials based on chitosan/cellulose composites have been prepared successfully with acetic acid [26] and trifluoroacetic acid [27] as solvents, respectively. Therefore, the excellent moisturizing properties of chitosan and its affinity with cellulose materials may enhance the environmental humidity tolerance and strength of cellulose-based electroactive materials.

Compared with lignocellulose, cotton fiber has a high content of α-cellulose, the degree of polymerization is relatively uniform, and the impurity content is very low, so it is often used as a raw material for the preparation of cellulose-based materials. In the study by Kim et al., due to the hydrophilicity of the cellulose and the water absorption of the ionic liquid, the osmotic pressure caused by the difference of concentration made the cellulose film doped with the ionic liquid absorb a large amount of water, and the internal ionic liquid gradually migrated to the surface of the material with the loss of water [18]. The high degree of deacetylation of the chitosan film is greatly compact, and it is not easy for water to permeate the membrane. For the chitosan/cellulose composite film, the loss of water in the material will be greatly reduced due to the presence of chitosan, which helps to ensure the migration of residual ionic liquid with free water in amorphous regions of the material, to a certain extent, and as a result, the durability and stability of the material are guaranteed [5,24,28]. 

Based on this, this paper will prepare cellulose/chitosan-based electroactive composite materials using an ionic liquid, 1-ethyl-3-methylimidazolium acetate ([EMIM]Ac), as solvent on the basis of our previous studies [29,30]. Chitosan/cellulose materials prepared with ionic liquids as solvents have a certain interpenetrating network structure, which has better strength and a denser surface [24,29]. Furthermore, the residual ionic liquids in the composite films can be controlled artificially in the preparation process, and then the electroactive behavior of the materials can be regulated. The residual ionic liquid [EMIM]Ac in the matrix has a strong adsorption of water [31], thus avoiding to some extent the weakening of the electroactive performance of materials caused by the gradual migration and disappearance of water molecules to the surface of materials. Finally, the EAP material prepared by poly(3,4-ethylenedioxythiophene)/poly (styrene sulfonate) (PEDOT:PSS) electrode not only exhibited good driving performance, but also ensured the stability and durability of the output of the driver [22].

## 2. Results and Discussion

### 2.1. FT-IR Analysis

It can be seen from Figure 1 that the absorption peak of N-H stretching vibration between 3100–3500 cm^-1^ coincided with the absorption peak of hydrogen bond O-H (3200–3500 cm^−1^) in the infrared spectrum [24], while the absorption peaks of -COOH were observed at 1650 cm^−1^ and 1402 cm^−1^, which were the characteristic peaks of the ionic liquid of [EMIM]Ac. Compared with the Ce-Ch-IL (Ce: Cellulose; Ch: Chitosan; IL: Ionic Liquid [EMIM]Ac) film, the -COOH characteristic peak of the Ce-IL film was more obvious, which implied that the ionic liquid content in the Ce-IL film was higher—that is, the ionic liquid was more likely to enter the Ce film than the Ce-Ch film.

### 2.2. XRD Analysis

X-ray diffractograms of Ce, Ce-IL, Ce-Ch, and Ce-Ch-IL films are shown in Figure 2. The peaks in the figure were 12.1°, 19.8°, and 22°, which correspond to (110), (110), and (200), respectively [32]. Among them, the diffraction overlap of (110) and (020) crystal planes formed a broad diffraction peak near 2θ = 22° [33,34], which confirmed that the cellulose crystal changed from cellulose type I to cellulose type II after cellulose and chitosan were dissolved and regenerated by [EMIM]Ac. Chitosan was positively charged while cellulose was negative charged, so it was easy to fabricate a uniform system because of the charge balance in the dissolution process. It can be clearly seen in Figure 2b that the crystallinity of the Ce-Ch film was higher than that of the Ce film. The highly deacetylated chitosan molecules were arranged very tightly to provide the high crystallinity of the chitosan/cellulose composite film [28,35,36]. The crystallinity of the Ce film increased after immersion in [EMIM]Ac due to the reordering of molecular chains during regeneration, and the crystallization peak of the Ce-Ch composite film did not change significantly, the reason possibly being that the Ce-Ch film was difficult to re-run due to the strong interaction between them, which was more conducive to reducing the sensitivity of the materials to environmental humidity. This result was consistent with the results of FTIR characterization. 

### 2.3. Thermogravimetric Analysis (TGA)

From Figure 3a, we can see that the mass loss of the films occurred at two stages. The first stage of the mass loss resulted from the evaporation of moisture trapped in cellulose and the second was due to the thermal decomposition of cellulose [37]. The maximum thermal decomposition temperature of cellulose was generally 340 °C [38], and the thermal decomposition temperature of the Ch film was about 270 °C to 280 °C. The thermal decomposition temperature of the Ce-Ch composite film was slightly lower than that of cellulose, indicating that the addition of a small amount of chitosan does not significantly reduce the thermal stability of cellulose. However, whether the cellulose or chitosan/cellulose composite film, when doped with ionic liquids, the thermal stability was significantly reduced due to the destruction of hydrogen bonds between the molecules. 

Figure 3b shows the water content of five kinds of films and the rate of loss of water contained in them. It can be clearly seen that the water content of the Ch film was much larger than that of the other films, with the Ch film showing excellent water retention performance. When cellulose and chitosan were combined to form a film, the water content of the composite film was also increased, and that is conducive to the maintenance of the material’s electroactive performance. 

Table 1 summarizes the data related to the water absorption and retention capacity of four films. It can be seen that the water absorption rate of the Ce-Ch film was decreased, but the water retention capacity was improved compared with the pure Ce film. This indicated that due to the dense arrangement of chitosan with high degree of deacetylation, the film formed a uniform and compact structure, which led to a decrease in the water exchange capacity between the film and the environment. However, the uniform distribution was more conducive to ion migration and moisture response. The water absorption rate and water retention capacity of the films are weakened after being doped with ionic liquids. 

### 2.4. SEM Analysis

Figure 4 shows SEM images of the surface and cross-sectional morphology of the Ce, Ce-Ch, and Ch films. It can be seen clearly that the surface of the Ce-Ch film is smoother and flatter than the pure Ce film and Ch film. This phenomenon demonstrated that the Ce-Ch film formed a denser structure on the surface, and the structure of a single component film was not so tight. This was also a reason why the change in the crystallinity of the Ce-Ch film after being doped with [EMIM]Ac was less than the pure Ce film, because the immersion in the [EMIM]Ac caused the further dissolution of the amorphous region of the fiber, and the crystallinity increased (which was consistent with the XRD spectrum), thereby causing the permeability of moisture and the high moisture retention area to reduce. Moreover, the variation of water content of the Ce-Ch film was lower than the pure Ce film after the immersion in [EMIM]Ac. This can be explained by the formation of hydrogen bonds between some hydroxyl groups in cellulose macromolecules and amino groups in chitosan molecules, and the formation of a compact composite structure, which weakened the water absorption properties of cellulose and was more conducive to reducing the environmental sensitivity and improving stability. 

As can be seen from Figure 4, compared with the smoother chitosan membrane, both the Ce and Ce-Ch films were multilayer structures due to the recrystallization of cellulose during the regeneration of cellulose from the ionic liquid. However, the internal fibers of the Ce film were aggregated, while the Ce-Ch film was more uniformly distributed, which was consistent with previous research results [28,39]. The phase inversion method by layer self-assembly was used to obtain a nano-scale multilayer structure, and more migratable holes were formed, which was beneficial for ion migration. The cations (NH_3_^+^) connected with chitosan molecular chains were fixed ions that cannot move freely, but the anions from [EMIM]Ac were free ions that can freely move [39]. Thus, under low DC voltage condition, the nano-scale multilayer structure accelerated the response to ions and moisture of the volume, and reduced the resistance [22,39].

There were many truncated fibers in the cross-section of the Ce film, as seen in Figure 4. This indicated that the fibers with lateral orientation were evenly distributed in the transverse direction. Piezoelectricity can be produced due to the single orientation alignment of the cellulose macromolecular crystals [3]. It was generally believed that the piezoelectric properties of cellulose itself and the internal ion migration process were the reasons for the electroactive behavior of cellulose materials [18]. Therefore, the rearrangement and recrystallization of cellulose during the preparation process led to improvements in crystallinity and orientation, which was beneficial to the electroactive properties of materials.

### 2.5. Mechanical Properties

Table 2 shows the data related to the mechanical properties of the films, including tensile strength and Young’s modulus calculated from the average value of five experimental trials. It can be seen from Table 2 that the tensile strength of the Ce-Ch film was 53.68% higher than that of the Ce film, and the Young’s modulus increased by 72.52%. This indicated that the doping of chitosan increased the interaction between cellulose and chitosan, and the mechanical properties were greatly improved. Due to the compatibility between natural cellulose and chitosan, cellulose-based materials can have stronger mechanical properties by forming complexes with chitosan [25]. Both the tensile strength and the Young’s modulus of the two films doped with the ionic liquid were lowered. It is well known that ionic liquids doped in cellulose can destroy some hydrogen bonds between cellulose molecules and weaken the interaction between fibers, leading the strength of the material to decrease significantly. However, the presence of ionic liquids provided more opportunities for ion migration and helped to improve the actuation properties of the film material.

### 2.6. Electroactive Performance

Figure 5 shows the relationship of the end bending deformation amplitudes of the four film materials with time at 10 V DC voltage. It was clearly observed that the electroactive displacement of the EAP material can be increased by doping chitosan and IL. The maximum end bending deformation amplitude of the Ce film was 2.2 mm, whereas the Ce-Ch film was 3.24 mm, having increased by 47.27%, and the Ce-IL film was 4.43 mm, having increased by 101.36%. This was because the crystallinity of the Ce-IL film increased after the immersion of IL, and the migration of residual ionic liquids under an electric field also contributed to their electrodynamic performance. However, the addition of chitosan in cellulose decreased the crystallinity of cellulose, resulted in the formation of a dense composite structure, and restricted the migration of some hydrated ions, so the sensitivity of the material deflection decreased significantly, although the maximum end bending deformation amplitude increased.

The Ce-Ch-IL film had the largest end bending deformation amplitude (7.3 mm), while it could also achieve a reverse deformation amplitude of 5.63 mm within 70 s when a reverse voltage was applied, thereby showing that it maintained a high reverse electroactive efficiency. However, the chitosan-cellulose composite EAPap actuator prepared by Kim et al. could achieve a bending deformation amplitude of 4.1 mm [24]. This comparison shows that the electroactive performance of EAP materials can be improved by impregnating ionic liquids. The end bending deformation amplitude of the Ce-Ch-IL film with time can be visually observed from Figure 5b. The fixed ions (NH_3_^+^) derived from chitosan macromolecules cannot move freely near to the negative electrode (cathode), while the free ions ([EMIM]Ac) move to positive electrode (anode). As the anions assemble at the anode, the repelling force between the anions makes the film bend to the negative electrode [39]. For the chitosan/cellulose composite film materials doped with ionic liquids, the addition of chitosan improved the water retention performance of the materials, and ionic liquids played a useful complement to the ion migration in the materials, so the materials showed good electroactive performance. Just as the SEM image shows that the Ce-Ch film has a denser surface structure and a more uniform internal system, although the crystallinity was slightly decreased after immersion in IL, the TG image shows that the final residual substance increased, which implies that the film has more IL remaining inside. These reasons all contributed to an increase in the electroactive displacement.

## 3. Experimental Method 

### 3.1. Materials

Cotton fiber was purchased from Xuzhou Health Material Factory Co., Ltd. (Jiangsu, China). Chitosan (N-deacetylation: ≥95%, crystallinity: 64.8%, viscosity: 100~200 mPa·s, biological reagent grade, PH: 7.0~9.0) was purchased from Aladdin Biotechnology Co., Ltd. (Shanghai, China). Ionic liquids (IL) of 1-ethyl-3-methylimidazolium acetate ([EMIM]Ac, 99%,) and poly(3,4-ethylenedioxythiophene)-poly (styrene sulfonate) (PEDOT: PSS, 1.5% in water) were purchased from Macklin Biochemical Co., Ltd. (Shanghai, China). All chemicals and materials were used as received, and all solutions were prepared using distilled water.

### 3.2. Fabrication of Electroactive Composites 

The cotton fiber and chitosan were placed in a vacuum oven at 50 °C for 24 h, and placed in a desiccator for constant weight. As shown in Figure 6, cotton fiber and chitosan with the ratio of 9:1 (w/w) were placed in a Teflon tube containing [EMIM]Ac ionic liquid with the solid content of 3%, and dissolved at low stirring speed for 3 h at 90 °C. Then, it was spin-coated at 600 r/min to form a film, which was precipitated in deionized water and soaked for 30 min, repeated three times, and dried in a vacuum oven at 50 °C for 3 h to obtain a regenerated chitosan/cellulose (Ce-Ch) film. Further, by the same method, a cellulose (Ce) film and a chitosan (Ch) film were prepared by dissolving 3% cotton fiber and 3% chitosan in [EMIM]Ac, respectively. Composite films doped with ionic liquids were obtained in the following ways: a Ce film and a Ce-Ch film were placed in 8% [EMIM]Ac aqueous solution for 1 h, and then dried in a vacuum oven at 50 °C for 1 h in order to acquire a Ce-IL film and a Ce-Ch-IL film, respectively. 

Four kinds of electroactive composites based on Ce, Ce-IL, Ce-Ch, and Ce-Ch-IL films were prepared by immersing them in PEDOT: PSS aqueous solution separately and were dried at 30 °C for 1 h. 

### 3.3. Electroactive Properties

The electroactive property testing device in this study is shown in Figure 7. The function signal generator and power amplifier were used as the source of the excitation electric field. The PXI (PCI extensions for instrumentation) digital-to-analog converter and the CCD (charge coupled device) camera were used to collect the electroactive deflection displacement. The LabVIEW was used as the visual motion analysis device to analyze the data under the excitation electric field. 

### 3.4. Structural Characterization of Materials

Fourier transform infrared spectrometer (FT-IR) of the films was analyzed using a VERTEX 70 (Bruker, Karlsruhe, Germany,) with a spectral range of 3500–500 cm^−1^. The crystalline structure was carried out on a Bruker D8 ADVANCE polycrystal X-ray diffraction (XRD, Karlsruhe, Germany) with 3 Kw output and a scanning range of 5° to 50° (2θ). Thermogravimetric analysis (TGA) was determined by using a simultaneous thermal analyzer (TG 209 F1 Libra, NETZSCH, Selb, Bavaria, Germany) at a heating rate of 10 °C/min from 30 °C to 700 °C under nitrogen atmosphere with a flow rate of 40 mL/min. The morphology of the surface and the cross-sections of the films was characterized by a Merlin scanning electron microscope (SEM, Zeiss, Oberkochen, Germany) at an accelerating voltage of 5 kV. 

### 3.5. Mechanical Characterization of Materials

The tensile strength and Young’s modulus were obtained by an INSTRON 5565 (INSTRON, Boston, Massachusetts, MA, USA) with a 100 N load cell at a cross-head speed of 0.1 mm/min.

### 3.6. Characterization of Water Absorption Rate and Water Retention Capacity

Water absorption rate and water retention capacity were evaluated using equations (1) [40] and (2) [41]. The films were cut into a shape with the dimensions of 2 cm × 2 cm, and the dry weight *m_0_* was weighed. Then the films were soaked in deionized water for 12 h, and the moisture on the surface was absorbed by filter paper, and the wet weight *m_1_* was recorded. Then the films were dried at 50 °C until the weight was constant, and the weight *m_2_* was recorded.

(1)$$\;\mathrm{Water}\;\mathrm{absorption}\;\mathrm{rate}\;=\frac{m_{1}-m_{0}}{m_{0}}\times 100\%$$

(2)$$\mathrm{Water}\;\mathrm{holding}\;\mathrm{capacity}=\frac{m_{2}}{m_{1}}\times 100\%$$

## 4. Conclusions

A kind of electroactive materials based on a chitosan/cellulose composite film was prepared successfully using PEDOT:PSS as conductive electrodes. The maximum end bending deformation amplitude of the resulted materials could reach up to 7.3 mm at 10 V DC voltage, which was 238.22% higher than that of the pure cellulose film, and could rapidly return to the initial position under the reverse DC voltage. In addition, the chitosan/cellulose composite film had lower water absorption, higher water retention, the tensile strength increased by 53%, and the Young’s modulus increased by 72.52%. SEM images indicated that the chitosan/cellulose composite film had a uniform internal distribution and a dense surface structure. 

## Figures and Tables

**Figure 1 ijms-20-06198-f001:**
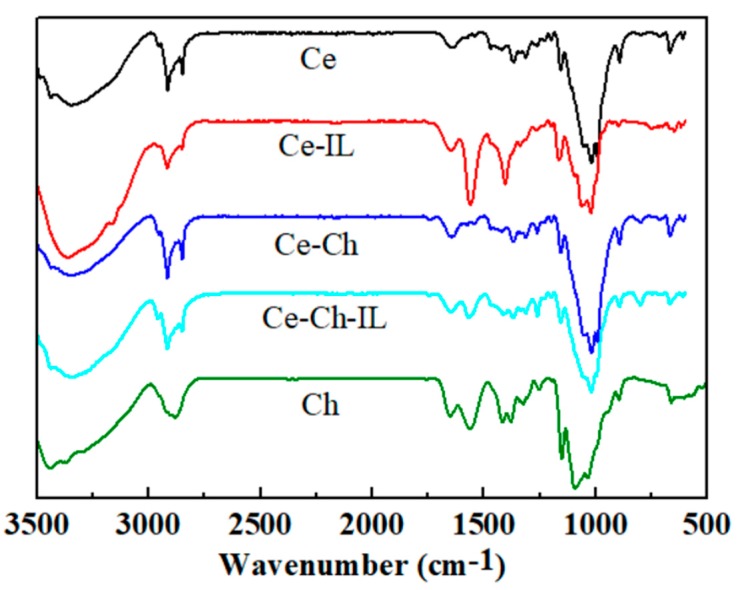
FT-IR spectra of Ce, Ce-IL, Ce-Ch, Ce-Ch-IL, and Ch films.

**Figure 2 ijms-20-06198-f002:**
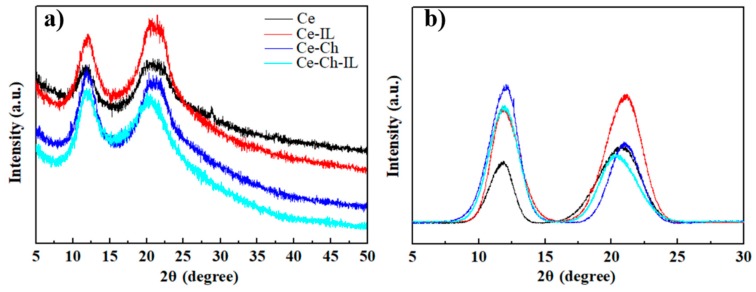
(**a**) X-ray diffraction patterns of Ce, Ce-IL, Ce-Ch, and Ce-Ch-IL films; (**b**) X-ray diffraction patterns of Ce, Ce-IL, Ce-Ch, and Ce-Ch-IL films with background curve editing.

**Figure 3 ijms-20-06198-f003:**
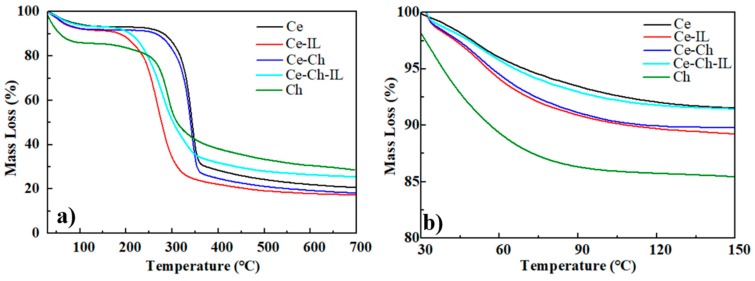
(**a**) Thermograms of Ce, Ce-IL, Ce-Ch, Ce-Ch-IL, and Ch films; (**b**) Thermograms of Ce, Ce-IL, Ce-Ch, Ce-Ch-IL, and Ch films at 30 °C to 150 °C.

**Figure 4 ijms-20-06198-f004:**
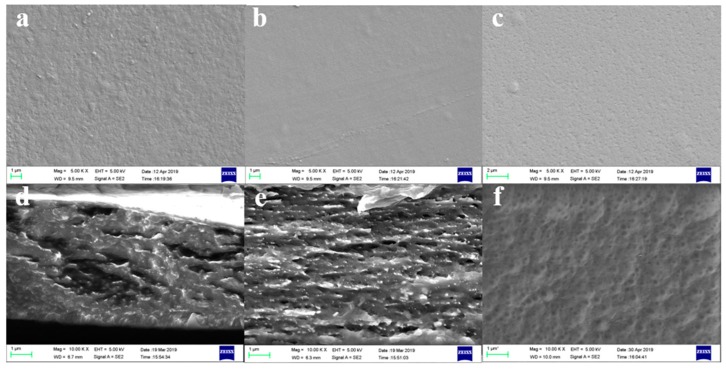
SEM images of samples: (**a**) Ce film, (**b**) Ce-Ch film, (**c**) Ch film, (**d**) Ce film cross-section, (**e**) Ce-Ch film cross-section, (**f**) Ch film cross-section.

**Figure 5 ijms-20-06198-f005:**
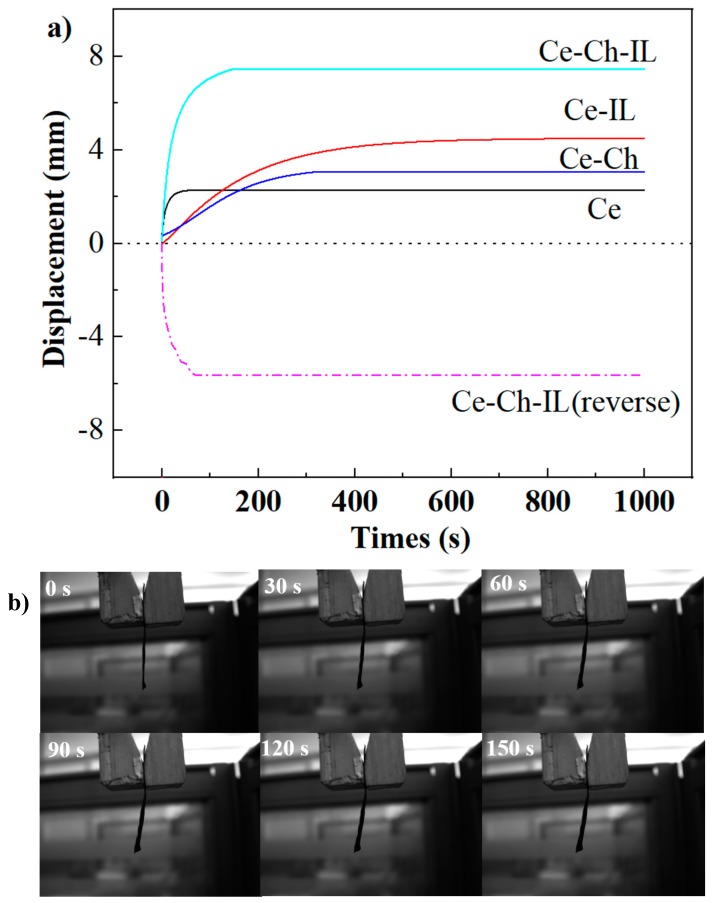
(**a**) Electroactive displacement of Ce, Ce-IL, Ce-Ch, and Ce-Ch-IL films with time at 10 V DC voltage; (**b**) real-time photos of Ce-Ch-IL deflection at different times.

**Figure 6 ijms-20-06198-f006:**
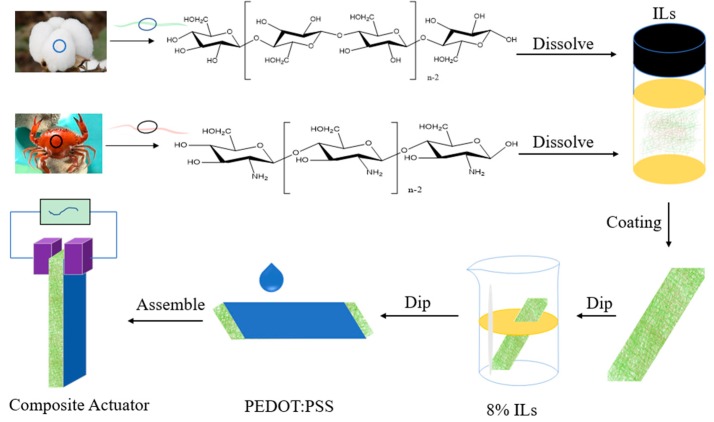
Schematic diagram of the preparation process of the electroactive composites.

**Figure 7 ijms-20-06198-f007:**
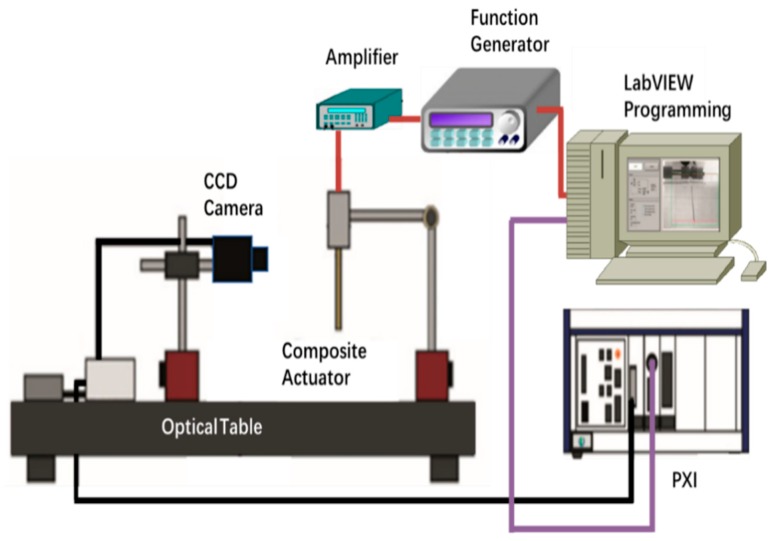
Schematic diagram of electroactive property testing device.

**Table 1 ijms-20-06198-t001:** The water absorption rate and water retention capacity of Ce, Ce-IL, Ce-Ch, and Ce-Ch-IL films.

Sample	Ce	Ce-Ch	Ce-IL	Ce-Ch-IL
water absorption rate (%)	47.50	25.00	40.91	16.67
water retention capacity (%)	64.41	86.67	58.06	82.86

**Table 2 ijms-20-06198-t002:** The tensile strength and Young’s modulus data of Ce, Ce-IL, Ce-Ch, and Ce-Ch-IL films.

Sample	Ce	Ce-Ch	Ce-IL	Ce-Ch-IL
Tensile strength (MPa)	32.90	50.56	24.51	38.21
Young’s modulus (MPa)	2968.50	5121.38	1919.49	2644.74

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
