# Peer review of "Study on the Preparation of Ionic Liquid Doped Chitosan/Cellulose-Based Electroactive Composites"

_ijms, 2019, doi:10.3390/ijms20246198_

Round 1
Reviewer 1 Report
The aim and the scientific thesis should be identified and conclusions section should be remodeled taking into the account the defined aim of the research and the thesis. The article should be remodeled in scheme as follows: introduction, materials, methods, results and discussion, conclusions.
More parameters of the chitosan: N-deacetylation, molecular weight, crystalinity. The origin of chitosan should be also described.
Water absorption rate and Water holding capacity are not coefficients connected to the structural properties. The methods of the above - mentioned coefficients determination should be described in the separated section of the methods. The same situation was found for the mechanical properties methods description.
Results presented in Table 2 should be given with the deviations.
The selection of the sources (chitosan and cotton) should be defined taking into the account the described state-of-the-art and the research thesis.
Author Response
Reviewer 1 Comment:
The aim and the scientific thesis should be identified and conclusions section should be remodeled taking into the account the defined aim of the research and the thesis. The article should be remodeled in scheme as follows: introduction, materials, methods, results and discussion, conclusions.Response: First of all, thank you for your constructive comments and suggestions. According to your opinions and suggestions, we revised and supplemented the paper in detail and seriously. We aimed to produce a cellulose-based electroactive material with high electroactive performance, good mechanical properties and low environmental sensitivity. The purpose and innovations of this article were emphasized in the abstract section, and the purpose of the paper was further explained in the introduction. Some scientific arguments for the effects of chitosan and ionic liquids on the performance of EAP materials were added. In addition, the conclusion part was revised and the structure of the article was remodeled as required by the journal.
More parameters of the chitosan: N-deacetylation, molecular weight, crystallinity. The origin of chitosan should be also described.
Response: Thank you for your reminding. We have supplemented the parameters of chitosan used in the experiment: N-deacetylation: ≥95%, crystallinity: 64.8%, viscosity: 100~200 mPa·s, biological reagent grade, PH:7.0~9.0, and it was purchased from aladdin Biotechnology Co., Ltd. (Shanghai, China).
Water absorption rate and Water holding capacity are not coefficients connected to the structural properties. The methods of the above mentioned coefficients determination should be described in the separated section of the methods. The same situation was found for the mechanical properties’ methods description.
Response: Your suggestion is very important. We have revised and improved the relevant content according to your opinion, separated the content of water absorption and water retention value from other content, and wrote it again.
Results presented in Table 2 should be given with the deviations.Response: Your suggestion is very important. We neglected it in the original. The average value of five measurements is listed in the table, which is supplemented in the revised version.
The selection of the sources (chitosan and cotton) should be defined taking into the account the described state-of-the-art and the research thesis.
Response: Thank you for your advice, we have modified and supplemented the relevant contents. Compared with lignocellulose, cotton fiber has a high content of α-cellulose, the degree of polymerization is relatively uniform, and the impurity content is very low, so it is often used as raw materials for the preparation of cellulose based materials. Chitosan has excellent water retention and biocompatibility, and it can provide a regular structure and high mechanical properties of the film.
Reviewer 2 Report
Although a lot of data for this study, there are some suggestions for the author.
1. In this study, the author must express purpose obviously for the study.
2. In part of the results and discussions, the author did not provide discussions to explain the experiment, also, did not give the reference to evident each experiment.
3. There was not a good organization for the manuscript. The author must review and reset it again.
Author Response
Reviewer 2 Comment:
Although a lot of data for this study, there are some suggestions for the author.
In this study, the author must express purpose obviously for the study.Response: Thank you for your constructive comments and suggestions. We revised the articles carefully, and rewrote some sentences, deleted and revised some inappropriate and misunderstand contents. We aimed to produce a cellulose-based electroactive material with high electroactive performance, good mechanical properties and low environmental sensitivity. First, the purpose of doping chitosan was to increase water retention and improve the adaptability of material to the environment. Second, the content of residual ions inside the material was increased by doping ionic liquids, and the end-group expansion effect of the material caused by ion migration was enhanced. These measures were beneficial to the electroactive performance of material. The purpose and innovations of this article were emphasized in the abstract section, and the purpose of the paper was further explained in the introduction.
In part of the results and discussions, the author did not provide discussions to explain the experiment, also, did not give the reference to evident each experiment.
Response: Your suggestion is very important. We discussed some of the discussions in more detail. XRD and SEM image analysis were partially supplemented and analyzed in line 111-113, 116-117 and 167-173. In addition, a more comprehensive analysis about the characterization of mechanical and electroactive performance were performed in line 190-192 and 212-218. We provided more references to support the above discussion.
There was not a good organization for the manuscript. The author must review and reset it again.
Response: First of all, thank you for your constructive comments and suggestions, we revised the articles carefully. The structure of the article was remodeled as required by the journal. In addition, we reorganized the article structure. The purpose of the paper was further elaborated in the introduction, the conclusion was revised, the results and discussion were further supplemented, and materials and methods were improved.
Round 2
Reviewer 1 Report
The article shows significant interest for the newly generated knowledge in range of the fabrication of the chitosan based materials using ionic liquids. The disadvantages of the previous version of the article were corrected by Authors.
Reviewer 2 Report
No comments for authors.